# Brains on Beats

**Umut Güçlü**
Radboud University, Donders Institute for
Brain, Cognition and Behaviour
Nijmegen, the Netherlands
u.guclu@donders.ru.nl

**Jordy Thielen**
Radboud University, Donders Institute for
Brain, Cognition and Behaviour
Nijmegen, the Netherlands
j.thielen@psych.ru.nl

**Michael Hanke**[*]
Otto-von-Guericke University Magdeburg
Center for Behavioral Brain Sciences
Magdeburg, Germany
michael.hanke@ovgu.de

**Marcel A. J. van Gerven**[†]
Radboud University, Donders Institute for
Brain, Cognition and Behaviour
Nijmegen, the Netherlands
m.vangerven@donders.ru.nl

## Abstract

We developed task-optimized deep neural networks (DNNs) that achieved state-of-the-art performance in different evaluation scenarios for automatic music tagging. These DNNs were subsequently used to probe the neural representations of music. Representational similarity analysis revealed the existence of a representational gradient across the superior temporal gyrus (STG). Anterior STG was shown to be more sensitive to low-level stimulus features encoded in shallow DNN layers whereas posterior STG was shown to be more sensitive to high-level stimulus features encoded in deep DNN layers.

## 1   Introduction

The human sensory system is devoted to the processing of sensory information to drive our perception of the environment [1]. Sensory cortices are thought to encode a hierarchy of ever more invariant representations of the environment [2]. A research question that is at the core of sensory neuroscience is what sensory information is processed as one traverses the sensory pathways from the primary sensory areas to higher sensory areas.

The majority of the work on auditory cortical representations has remained limited to understanding the neural representation of hand-designed low-level stimulus features such as spectro-temporal models [3], spectro-location models [4], timbre, rhythm, tonality [5–7] and pitch [8] or high-level representations such as music genre [9] and sound categories [10]. For example, Santoro et al. [3] found that a joint frequency-specific modulation transfer function predicted observed fMRI activity best compared to frequency-nonspecific and independent models. They showed specificity to fine spectral modulations along Heschl's gyrus (HG) and anterior superior temporal gyrus (STG), whereas coarse spectral modulations were mostly located posterior-laterally to HG, on the planum temporale (PT), and STG. Preference for slow temporal modulations was found along HG and STG, whereas fast temporal modulations were observed on PT, and posterior and medially adjacent to HG. Also, it has been shown that activity in STG, somatosensory cortex, the default mode network, and cerebellum are sensitive to timbre, while amygdala, hippocampus and insula are more sensitive to rhythmic and

---

[*]http://psychoinformatics.de; supported by the German federal state of Saxony-Anhalt and the European Regional Development Fund (ERDF), project: Center for Behavioral Brain Sciences.

[†]http://www.ccnlab.net; supported by VIDI grant 639.072.513 of the Netherlands Organization for Scientific Research (NWO).

tonality features [5, 7]. However these efforts have not yet provided a complete algorithmic account of sensory processing in the auditory system.

Since their resurgence, deep neural networks (DNNs) coupled with functional magnetic resonance imaging (fMRI) have provided a powerful approach to form and test alternative hypotheses about what sensory information is processed in different brain regions. On one hand, a task-optimized DNN model learns a hierarchy of nonlinear transformations in a supervised manner with the objective of solving a particular task. On the other hand, fMRI measures local changes in blood-oxygen-level dependent hemodynamic responses to sensory stimulation. Subsequently, any subset of the DNN representations that emerge from this hierarchy of nonlinear transformations can be used to probe neural representations by comparing DNN and fMRI responses to the same sensory stimuli. Considering that the sensory systems are biological neural networks that routinely perform the same tasks as their artificial counterparts, it is not inconceivable that DNN representations are suitable for probing neural representations.

Indeed, this approach has been shown to be extremely successful in visual neuroscience. To date, several task-optimized DNN models were used to accurately model visual areas on the dorsal and ventral streams [11–18], revealing representational gradients where deeper neural network layers map to more downstream areas along the visual pathways [19, 20]. Recently, [21] has shown that deep neural networks trained to map speech excerpts to word labels could be used to predict brain responses to natural sounds. Here, deeper neural network layers were shown to map to auditory brain regions that were more distant from primary auditory cortex.

In the present work we expand on this line of research where our aim was to model how the human brain responds to music. We achieve this by probing neural representations of music features across the superior temporal gyrus using a deep neural network optimized for music tag prediction. We used the representations that emerged after training a DNN to predict tags of musical excerpts as candidate representations for different areas of STG in representational similarity analysis. We show that different DNN layers correspond to different locations along STG such that anterior STG is shown to be more sensitive to low-level stimulus features encoded in shallow DNN layers whereas posterior STG is shown to be more sensitive to high-level stimulus features encoded in deep DNN layers.

## 2    Materials and Methods

### 2.1    *MagnaTagATune* Dataset

We used the MagnaTagATune dataset [22] for DNN estimation. The dataset contains 25.863 music clips. Each clip is a 29 seconds long excerpt from 5223 songs from 445 albums from 230 artists. Each excerpt is supplied with a vector of binary annotations of 188 tags. These annotations are obtained by humans playing the two-player online TagATune game. In this game, the two players are either presented with the same or a different audio clip. Subsequently, they are asked to come up with tags for their specific audio clip. Afterward, players view each other's tags and are asked to decide whether they were presented the same audio clip. Tags are only assigned when more than two players agreed. The annotations include tags like 'singer', 'no singer', 'violin', 'drums', 'classical', 'jazz', et cetera. We restricted our analysis on this dataset to the top 50 most popular tags to ensure that there is enough training data for each tag. Parts 1-12 were used for training, part 13 was used for validation and parts 14-16 were used for testing.

### 2.2    *Studyforrest* Dataset

We used the existing studyforrest dataset [23] for representational similarity analysis. The dataset contains fMRI data on the perception of musical genres. Twenty participants (age 21-38 years, mean age 26.6 years), with normal hearing and no known history of neurological disorders, listened to twenty-five 6 second, 44.1 kHz music clips. The stimulus set comprised five clips per each of the five following genres: Ambient, Roots Country, Heavy Metal, 50s Rock 'n Roll, and Symphonic. Stimuli were selected according to the procedure of [9]. The Ambient and Symphonic genres can be considered as non-vocal and the others as vocal. Participants completed eight runs, each with all twenty-five clips.

Ultra-high-field (7 Tesla) fMRI images were collected using a Siemens MAGNETOM scanner, T2*-weighted echo-planar images (gradient-echo, repetition time (TR) = 2000 ms, echo time (TE) = 22 ms, 0.78 ms echo spacing, 1488 Hz/Px bandwidth, generalized auto-calibrating partially parallel acquisition (GRAPPA), acceleration factor 3, 24 Hz/Px bandwidth in phase encoding direction), and a 32 channel brain receiver coil. Thirty-six axial slices were acquired (thickness = 1.4 mm, 1.4 × 1.4 mm in-plane resolution, 224 mm field-of-view (FOV) centered on the approximate location of Heschl's gyrus, anterior-to-posterior phase encoding direction, 10% inter-slice gap). Along with the functional data, cardiac and respiratory traces, and a structural MRI were collected. In our analyses, we only used the data from the 12 subjects (Subjects 1, 3, 4, 6, 7, 9, 12, 14–18) with no known data anomalies as reported in [23].

The anatomical and functional scans were preprocessed as follows: Functional scans were realigned to the first scan of the first run and next to the mean scan. Anatomical scans were coregistered to the mean functional scan. Realigned functional scans were slice-time corrected to correct for the differences in image acquisition times between the slices. Realigned and slice-time corrected functional scans were normalized to MNI space. Finally, a general linear model was used to remove noise regressors derived from voxels unrelated to the experimental paradigm and estimate BOLD response amplitudes [24]. We restricted our analyses to the superior temporal gyrus (STG).

## 2.3 Deep Neural Networks

We developed three task-optimized DNN models for tag prediction. Two of the models comprised five convolutional layers followed by three fully-connected layers (DNN-T model and DNN-F model). The inputs to the models were 96000-dimensional time (DNN-T model) and frequency (DNN-F model) domain representations of six second-long audio signals, respectively. One of the models comprised two streams of five convolutional layers followed by three fully connected layers (DNN-TF model). The inputs to the streams were given by the time and frequency representations. The outputs of the convolutional streams were merged and fed into first fully-connected layer. Figure 1 illustrates the architecture of the one-stream models.

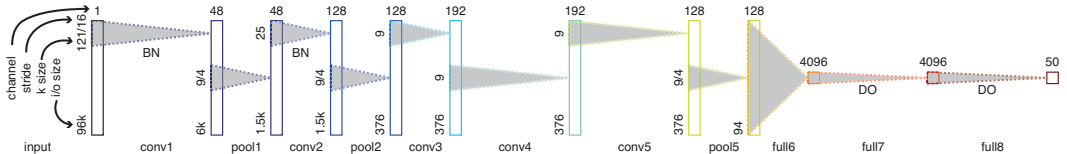

Figure 1: **Architecture of the one-stream models.** First seven layers are followed by parametric softplus units [25], and the last layer is followed by sigmoid units. The architecture is similar to that of AlexNet [26] except for the following modifications: (i) The number of convolutional kernels are halved. (ii) The (convolutional and pooling) kernels and strides are flattened. That is, an $n \times n$ kernel is changed to an $n^2 \times 1$ kernel and an $m \times m$ stride is changed to an $m^2 \times 1$ stride. (iii) Local response normalization is replaced with batch normalization [27]. (iv) Rectified linear units are replaced with parametric softplus units with initial $\alpha = 0.2$ and initial $\beta = 0.5$. (v) Softmax units are replaced with sigmoid units.

We used Adam [28] with parameters $\alpha = 0.0002$, $\beta_1 = 0.5$, $\beta_2 = 0.999$, $\epsilon = 1e^{-8}$ and a mini batch size of 36 to train the models by minimizing the binary cross-entropy loss function. Initial model parameters were drawn from a uniform distribution as described in [29]. Songs in each training mini-batch were randomly cropped to six seconds (96000 samples). The epoch in which the validation performance was the highest was taken as the final model (53, 12 and 12 for T, F and TF models, respectively). The DNN models were implemented in Keras [30].

Once trained, we first tested the tag prediction performance of the models and identified the model with the highest performance. To predict the tags of a 29 second long song excerpt in the test split of the MagnaTagaTune dataset, we first predicted the tags of 24 six-second-long overlapping segments separated by a second and averaged the predictions.

We then used the model with the highest performance for nonlinearly transforming the stimuli to eight layers of hierarchical representations for subsequent analyses. Note that the artificial neurons in the convolutional layers locally filtered their inputs (1D convolution), nonlinearly transformed them

and returned temporal representations per stimulus. These representations were further processed by averaging them over time. In contrast, the artificial neurons in the fully-connected layers globally filtered their inputs (dot product), non-linearly transformed them and returned scalar representations per stimulus. These representations were not further processed. These transformations resulted in $n$ matrices of size $m \times p_i$ where $n$ is the number of layers (8), $m$ is the number of stimuli (25) and $p_i$ is the number of artificial neurons in the $i$th layer (48 or 96, 128 or 256, 192 or 384, 192 or 384, 128 or 256, 4096, 4096 and 50 for $i = 1, \ldots, 8$, respectively).

## 2.4   Representational Similarity Analysis

We used Representational Similarity Analysis (RSA) [31] to investigate how well the representational structures of DNN model layers match with that of the response patterns in STG. In RSA, models and brain regions are characterized by $n \times n$ representational dissimilarity matrices (RDMs), whose elements represent the dissimilarity between the neural or model representations of a pair of stimuli. In turn, computing the overlap between the model and neural RDMs provides evidence about how well a particular model explains the response patterns in a particular brain region. Specifically, we performed a region of interest analysis as well as a searchlight analysis by first constructing the RDMs of STG (target RDM) and the model layers (candidate RDM). In the ROI analysis, this resulted in one target RDM per subject and eight candidate RDMs. For each subject, we correlated the upper triangular parts of the target RDM with the candidate RDMs (Spearman correlation). We quantified the similarity of STG representations with the model representations as the mean correlation. For the searchlight analysis, this resulted in 27277 target RDMs (each derived from a spherical neighborhood of 100 voxels) and 8 candidate RDMs. For each subject and target RDM, we correlated the upper triangular parts of the target RDM with the candidate RDMs (Spearman correlation). Then, the layers which resulted in the highest correlation were assigned to the voxels at the center of the corresponding neighborhoods. Finally, the layer assignments were averaged over the subjects and the result was taken as the final layer assignment of the voxels.

## 2.5   Control Models

To evaluate the importance of task optimization for modeling STG representations, we compared the representational similarities of the entire STG region and the task-optimized DNN-TF model layers with the representational similarities of the entire STG region and two sets of control models.

The first set of control models transformed the stimuli to the following 48-dimensional model representations[3]:

- Mel-frequency spectrum (mfs) representing a mel-scaled short-term power spectrum inspired by human auditory perception where frequencies organized by equidistant pitch locations. These representations were computed by applying (i) a short-time Fourier transform and (ii) a mel-scaled frequency-domain filterbank.

- Mel-frequency cepstral coefficients (mfccs) representing both broad-spectrum information (timbre) and fine-scale spectral structure (pitch). These representations were computed by (i) mapping the mfs to a decibel amplitude scale and (ii) multiplying them by the discrete cosine transform matrix.

- Low-quefrency mel-frequency spectrum (lq_mfs) representing timbre. These representations were computed by (i) zeroing the high-quefrency mfccs, (ii) multiplying them by the inverse of discrete cosine transform matrix and (iii) mapping them back from the decibel amplitude scale.

- High-quefrency mel-frequency spectrum (hq_mfs) representing pitch. These representations were computed by (i) zeroing the low-quefrency mfccs, (ii) multiplying them by the inverse of discrete cosine transform matrix and (iii) mapping them back from the decibel amplitude scale.

The second set of control models were 10 random DNN models with the same architecture as the DNN-TF model, but with parameters drawn from a zero mean and unit variance multivariate Gaussian distribution.

## 3 Results

In the first set of experiments, we analyzed the task-optimized DNN models. The tag prediction performance of the models for the individual tags was defined as the area under the receiver operator characteristics (ROC) curve (AUC).

We first compared the mean performance of the models over all tags (Figure 2). The performance of all models was significantly above chance level ($p \ll 0.001$, Student's $t$-test, Bonferroni correction). The highest performance was achieved by the DNN-TF model (0.8939), followed by the DNN-F model (0.8905) and the DNN-T model (0.8852). To the best of our knowledge, this is the highest tag prediction performance of an *end-to-end* model evaluated on the same split of the same dataset [32]. The performance was further improved by averaging the predictions of the DNN-T and DNN-F models (0.8982) as well as those of the DNN-T, DNN-F and DNN-TF models (0.9007). To the best of our knowledge, this is the highest tag prediction performance of *any* model (ensemble) evaluated on the same split of the same dataset [33, 32, 34]. For the remainder of the analyses, we considered only the DNN-TF model since it achieved the highest single-model performance.

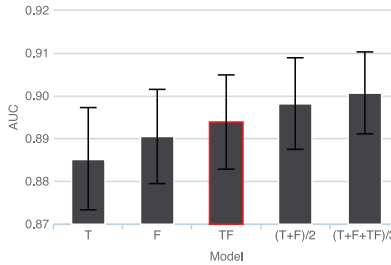

Figure 2: **Tag prediction performance of the task-optimized DNN models.** Bars show AUCs over all tags for the corresponding task-optimized DNN models. Error bars show $\pm$ SE. All pairwise differences are significant except for the pairs 1 and 2, and 2 and 3 ($p < 0.05$, paired-sample $t$-test, Bonferroni correction).

We then compared the performance of the DNN-TF model for the individual tags (Figure 3). Visual inspection did not reveal a prominent pattern in the performance distribution over tags. The performance was not significantly correlated with tag popularity ($p > 0.05$, Student's $t$-test). The only exception was that the performance for the positive tags were significantly higher than that for the negative tags ($p \ll 0.001$, Student's $t$-test).

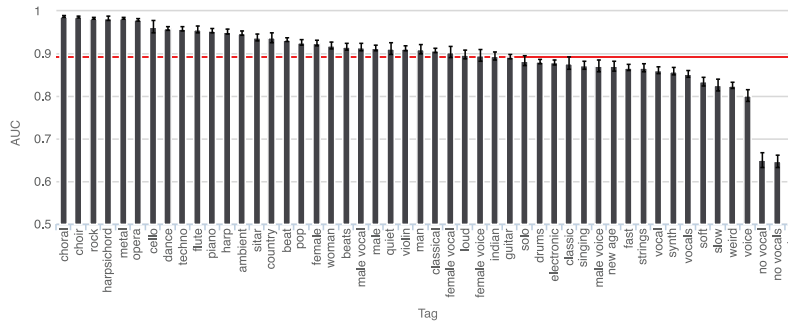

Figure 3: **Tag prediction performance of the task-optimized DNN-TF model.** Bars show AUCs for the corresponding tags. Red band shows the mean $\pm$ SE for the task-optimized DNN-TF model over all tags.

In the second set of experiments, we analyzed how closely the representational geometry of STG is related to the representational geometries of the task-optimized DNN-TF model layers.

First, we constructed the candidate RDMs of the layers (Figure 4). Visual inspection revealed similarity structure patterns that became increasingly prominent with increasing layer depth. The most prominent pattern was the non-vocal and vocal subdivision.

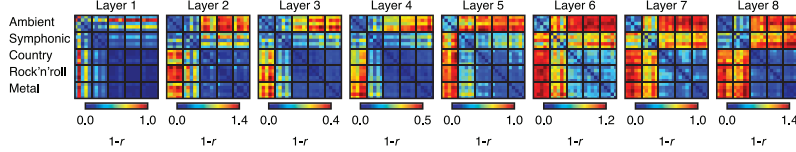

Figure 4: **RDMs of the task-optimized DNN-TF model layers.** Matrix elements show the dissimilarity (1 - Spearman's $r$) between the model layer representations of the corresponding trials. Matrix rows and columns are sorted according to the genres of the corresponding trials.

Second, we performed a region of interest analysis by comparing the reference RDM of the entire STG region with the candidate RDMs (Figure 5). While none of the correlations between the reference RDM and the candidate RDMs reached the noise ceiling (expected correlation between the reference RDM and the RDM of the true model given the noise in the analyzed data [31]), they were all significantly above chance level ($p < 0.05$, signed-rank test with subject RFX, FDR correction). The highest correlation was found for Layer 1 (0.6811), whereas the lowest correlation was found for Layer 8 (0.4429).

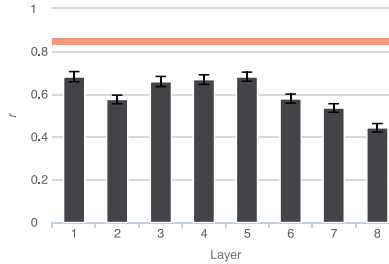

Figure 5: **Representational similarities of the entire STG region and the task-optimized DNN-TF model layers.** Bars show the mean similarity (Spearman's $r$) of the target RDM and the corresponding candidate RDMs over all subjects. Error bars show $\pm$ SE. Red band shows the expected representational similarity of the STG and the true model given the noise in the analyzed data (noise ceiling). All pairwise differences are significant except for the pairs 1 and 5, 2 and 6, and 3 and 4 ($p < 0.05$, signed-rank test with subject RFX, FDR correction).

Third, we performed a searchlight analysis [35] by comparing the reference RDMs of multiple STG voxel neighborhoods with the candidate RDMs (Figure 6). Each neighborhood center was assigned a layer such that the corresponding target and candidate RDM were maximally correlated. This analysis revealed a systematic change in the mean layer assignments over subjects along STG. They increased from anterior STG to posterior STG such that most voxels in the region of the transverse temporal gyrus were assigned to the shallower layers and most voxels in the region of the angular gyrus were assigned to the deeper layers. The corresponding mean correlations between the target and the candidate RDMs decreased from anterior to posterior STG.

In order to quantify the gradient in layer assignment, we correlated the mean layer assignment of the STG voxels in each coronal slice with the slice position, which was taken to be the slice number. As a result, it was found that layer and position are significantly correlated for the voxels along the anterior - posterior STG direction ($r = 0.7255$, Pearson's $r$, $p \ll 0.001$, Student's $t$-test). Furthermore, the mean correlations between the target and the candidate RDMs for the majority (85.53%) of the STG voxels were significant ($p < 0.05$, signed-rank test with subject RFX, FDR correction for the number of voxels followed by Bonferroni correction for the number of layers). However, the correlations of many voxels at the posterior end of STG were not highly significant in contrast to their central counterparts and ceased to be significant as the (multiple comparisons corrected) critical value was decreased from 0.05 to 0.01, which reduced the number of voxels surviving the critical value from 85.53% to 75.32%. Nevertheless, the gradient in layer assignment was maintained even when the voxels that did not survive the new critical value were ignored ($r = 0.7332$, Pearson's $r$, $p \ll 0.001$, Student's $t$-test).

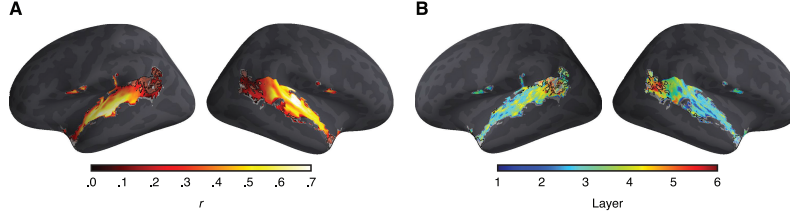

Figure 6: **Representational similarities of the spherical STG voxel clusters and the task-optimized DNN-TF model layers.** Only the STG voxels that survived the (multiple comparisons corrected) critial value of 0.05 are shown. Those that did not survive the critical value of 0.01 are indicated with transparent white masks and black outlines. (**A**) Mean representational similarities over subjects. (**B**) Mean layer assignments over subjects.

These results show that increasingly posterior STG voxels can be modeled with increasingly deeper DNN layers optimized for music tag prediction. This observation is in line with the visual neuroscience literature where it was shown that increasingly deeper layers of DNNs optimized for visual object and action recognition can be used to model increasingly downstream ventral and dorsal stream voxels [19, 20]. It also agrees with previous work showing a gradient in auditory cortex with DNNs optimized for speech-to-word mapping [21]. It would be of particular interest to compare the respective gradients and use the music and speech DNNs as each other's control model such as to disentangle speech- and music-specific representations in auditory cortex.

In the last set of experiments, we analyzed the control models. We first constructed the RDMs of the control models (Figure 7). Visual inspection revealed considerable differences between the RDMs of the task-optimized DNN-TF model and those of the control models.

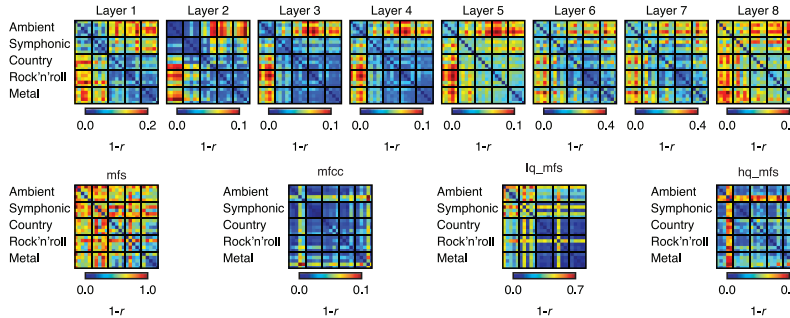

Figure 7: **RDMs of the random DNN model layers (top row) and the baseline models (bottom row).** Matrix elements show the dissimilarity (1 - Spearman's $r$) between the model layer representations of the corresponding trials. Matrix rows and columns are sorted according to the genres of the corresponding trials.

We then compared the similarities of the task-optimized candidate RDMs and the target RDM versus the similarities of the control RDMs and the target RDM (Figure 8). The layers of the task-optimized DNN model significantly outperformed the corresponding layers of the random DNN model ($\Delta r = 0.21$, $p < 0.05$, signed-rank test with subject RFX, FDR correction) and the four baseline models ($\Delta r = 0.42$ for mfs, $\Delta r = 0.21$ for mfcc, $\Delta r = 0.44$ for lq_mfs and $\Delta r = 0.34$ for hq_mfs, signed-rank test with subject RFX, FDR correction). Furthermore, we performed the searchlight analysis with the random DNN model to determine whether the gradient in layer assignment is a consequence of model architecture or model representation. We found that the random DNN model failed to maintain the gradient in layer assignment ($r = -0.2175$, Pearson's $r$, $p = 0.0771$, Student's $t$-test), suggesting that the gradient is in the representation that emerges from task optimization.

These results show the importance of task optimization for modeling STG representations. This observation also is line with visual neuroscience literature where similar analyses showed the importance of task optimization for modeling ventral stream representations [19, 17].

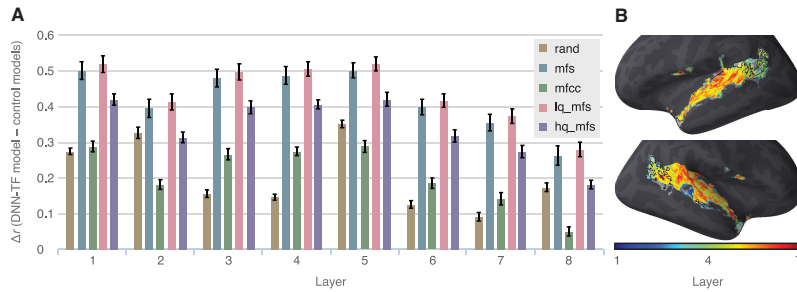

Figure 8: **Control analyses.** (**A**) Representational similarities of the entire STG region and the task-optimized DNN-TF model versus the representational similarities of the entire STG region and the control models. Different colors show different control models: Random DNN model, mfs model, mfcc model, lq_mfs model and hq_mfs model. Bars show mean similarity differences over subjects. Error bars show $\pm$ SE. (**B**) Mean layer assignments over subjects for the random DNN model. Voxels, masks and outlines are the same as those in Figure 6.

## 4   Conclusion

We showed that task-optimized DNNs that use time and/or frequency domain representations of music achieved state-of-the-art performance in various evaluation scenarios for automatic music tagging. Comparison of DNN and STG representations revealed a representational gradient in STG with anterior STG being more sensitive to low-level stimulus features (shallow DNN layers) and posterior STG being more sensitive to high-level stimulus features (deep DNN layers). These results, in conjunction with previous results on the visual and auditory cortical representations, suggest the existence of multiple representational gradients that process increasingly complex conceptual information as we traverse sensory pathways of the human brain.

## Footnotes

[3]These are provided as part of the *studyforrest* dataset [23].

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
