[Reviews · NeurIPS 2016]

Reviewer 1

Summary

The authors train a deep neural (8 layer) network to assign tags to music clips. They summarize the representation in each layer by constructing a dissimilarity matrix across different musical tags (i.e each entry in the matrix is the 1 - the correlation between the activations for each tag). This representational summary is compared to a that of fMRI data collected from the superior temporal gyrus (SG). The authors report a representational gradient along the SG which correlates with depth in the network. This suggests that there is a gradient along the SG giving rise to increasingly complex representations of auditory inputs.

Qualitative Assessment

This is an interesting study, following a line of studies in the visual system trying to link the representation of stimuli in different layers of artificial neural networks to the representation in different stages of biological neural processing. The authors claim (and I do not dispute this claim) That this is the first such study in the auditory system, making this study novel and potentially impactful. My main concern is the dimensionality of the comparisons, and in particular the searchlight approach. The images in Fig. 5 are indeed compelling, but they have the potential to be misleading. First, us humans tend to look for patterns in images, so it is important to provide more objective summaries. Second, as seen in Fig. 5A, not all regions are correlated to the same extent. It is important to take this into account. In particular, it is important to avoid summing over regions that are not significantly correlated (and beware of multiple hypothesis testing when judging whether a voxel is significantly correlated). Thus, I suggest that the authors provide a quantification of the relationship between layer depth and position along the SG. For example, pick a direction along the SG and correlate it with mean layer depth. Ideally such a quantification will also be presented with a measure of confidence (some kind of error bar) to verify the effects are real. Another important question is how do theses results relate to current knowledge about the SG? Are there previous reports/physiology data indicating a representational gradient in SG or is this completely novel? As a reader who is not familiar with this literature, I think it is important to discuss these details. More minor comments: 1. Methods - I was a bit confused whether the authors conducted new fMRI experiments or just used an existing dataset. They mention using the studyforrest dataset, but then the methods description gives the impression that experiments were conducted as part of this study. It would be nice to clarify this point. 2. In Fig. 3, why not use the same color scale for all plots? Also isn't 1-r bounded between 1 and -1? If so why are the values outside this bound? Why are the values only positive? Does this mean that all the Spearman correlations are negative?

Confidence in this Review

2-Confident (read it all; understood it all reasonably well)


Reviewer 2

Summary

The paper trains three different deep neural networks on a large repertoire of tagged music samples and reports new state-of-the-art performance for tag prediction. The best performing network is then fed song clips which were used in an fMRI study with human participants and activation in each of the layers of the network is compared to the average brain activity recorded from the human subjects via representational similarity analysis (RSA). The paper reports that all DNN layers demonstrated above chance similarity with brain signals recorded in the superior temporal gyrus (STG) and finds that as you move anterior to posterior, activation of shallow and then deep DNN layers are best correlated with the brain data. The paper also reports that similarity between brain activity and control models (randomly initialized DNNs or models which extract basic audio features from the data) show less similarity with brain data than do the optimized DNNs initially trained on the music samples.

Qualitative Assessment

The paper is extremely clear and easy to read. The two contributions of the paper are (1) DNNs which achieve new state-of-the-art performance for music tag prediction and (2) the finding of a gradient along the STG in which anterior brain activity is more similar to the activation of shallow layers of a DNN and posterior activity is more similar to the activation of deeper layers of a DNN. The achievement of state-of-the-art music tagging performance is notable. The general approach of relating the activity of layers of a DNN to brain activity is not novel to this paper and there are recent publications in the visual system which employ this approach (as cited by the authors). From a neuroscience perspective, the contribution of the paper must therefore lie in it's scientific findings - which seems to be the finding of the gradient noted above. The paper does not attempt to provide a description of the features represented by the different layers of the network, which would be helpful in getting a sense of which information is processed by the brain as you move along the gradient the authors have found. Without this, we are left to assume that as deeper layers of a DNN represent more complex features this same principle may apply along the STG, but interpretation beyond this is not possible based on what is presented in this paper. While it would be gratifying to be able to more deeply interpret what it being processed along the gradient the authors report, I believe the basic finding of the gradient itself coupled with the state-of-the-art music tagging would likely be of at least mild interest and useful to a broad audience, which has motivated my scores above.

Confidence in this Review

2-Confident (read it all; understood it all reasonably well)


Reviewer 3

Summary

This paper has two parts. First, they train a DNN to classify music-related tags from the Magnatagatune dataset of music audio. They then relate the structure of the network to the brain, specifically the Superior Temporal Gyrus (STG). They show that the representations learned by the DNN can be mapped onto the STG in the form of a "representational gradient." In other words (from the abstract, in fact) "low-level stimulus features encoded in shallow DNN layers whereas posterior STG was shown to be more sensitive to high-level stimulus features encoded in deep DNN layers."

Qualitative Assessment

I'm pretty familiar with the music side of this work, including the DNN and the use of Magnatagatune. I thought the writeup was clear and concise, and the ML work was good. No complaints. I'm much less familiar with the neuroscience part of the paper. From my viewpoint, it's an amazing connection to draw, and I loved reading about it. My concern is that it's overfitting in some way that I don't understand. But that's just the natural response, given that I'm a reviewer with no real training in these methods. Overall I loved this paper. I'll highly recommend that my colleagues read it. I strongly believe it should be accepted and think it would make a great Oral.

Confidence in this Review

2-Confident (read it all; understood it all reasonably well)


Reviewer 4

Summary

The authors try to confirm the theoretical intuition, recently developed around the Visual Cortex, that the sample-kernel space induced by the representation coded by a neural network trained for music tagging and those of the auditory cortex exposed to the same stimuli are partially aligned. Furthermore they investigate the presence of a common complexity gradient emerging layerwise for the CNN and through the anterior-posterior direction in the auditory cortex. They use representational similarity analysis, 1-spearman correlation of the sample kernel matrix, to measure the aligment between a CNN and a human brain exposed to the same stimuli. The brain data used come from a high quality open access dataset.

Qualitative Assessment

The article extend the recent CNN-Human Brain alignment literature to auditory data, but fails to do so in a statistically principled way. While the construction of the artificial model and the collection of the data are sound, their comparison is vague and potentially exhibiting some false positive results, especially with respect to the claim of the presence of an unsupervised ---> goal directed complexity gradient across the superior temporal gyrus. While this type of analysis might be accepted in Neuroscience journals the NIPS community has been producing fundamental results on the alignment of high dimensional spaces that if integrated in the current analysis would allow to confirm whether the results are false positives or they are sound.Results are statistically weak and tested with a not enough rigorous procedure. Two of the most complex dynamical systems we currently have access to (Human Brains and Convolutional Neural Neworks) are given access to the same stimuli and the representational spaces they induce are compared by a simple Spearman Correlation measure. Potential flaws in this procedure (that is more than common in Neuroscience studies) had been previously exposed in https://hal.inria.fr/hal-01187297/file/paper_stamlins.pdf (Correlations of correlations are not reliable statistics) . Figure 1 is in a 0.05 AUC range potentially showing non-existing difference in fitting of the different models. In figure 4 there seems to be no real difference across layers. Figure 5 presents a graphical illustration after the searchlight procedure that seems really prone to produce a strong selection bias unless a (non-described) cross validation procedure is introduced. Figure 7 shows that the neural networks with random weights has the smaller (0.2) difference with respect to the proposed fine-tuned architecture, it would be interesting if the authors applied their searchlight procedure to this architecture as it would most likely convey some statistically significant results. With respect to the comparison considering each kernel matrix as a graph and embedding in a new space with a graph kernel could be a direction, simply using a measure respecting the manifold of covariance matrices would be already a good improvement. It seems odd the authors did not use the cross-validated mass ridge regression framework that has been largely used in previous similar articles.

Confidence in this Review

3-Expert (read the paper in detail, know the area, quite certain of my opinion)


Reviewer 5

Summary

This paper performs a RSA similarity analysis assessment of the similarity of trained DNN to human fMRI traces, using music samples as stimuli. This type of approach has been highly successful in the visual domain to improve on the understanding of the visual hierarchy, finding great correlations between DNN trained on ImageNet and human/monkey IT. The authors replicate this using music datasets (MagnaTagaTune to train the DNN, Studyforrest for the RSA). Their DNN models trained on MagnaTagaTune obtain SOTA classification accuracy. They then find that DNN layers representations similarity seems to go from anterior to posterior along the STG, providing some hints about the salient features represented in STG.

Qualitative Assessment

This paper has 2 main contributions: 1) Improve the SOTA on DNN for music tag classification. 2) Look for correlations with the human STG through a RSA + Searchlight procedure. The treatment of 1) is good and well explained. In Section 2.3, it would be more helpful to spell out directly the number of filters, along with sizes and strides per layers, instead of referring to AlexNet and providing the set of rules to adapt them... A small figure would be more space efficient. All the latest tricks and techniques were used, there were no flawed in my opinion. Compared to the previous state of the art in music classification using DNN, it seems like the main difference is a drastic change in filter sizes and using the latest optimizers. If I parsed the methods properly, they use conv filter sizes {121, 25, 9, 9, 9} (strides similarly adapted), compared to {8, 8} in [32]. This is quite a drastic improvement, which might explain the SOTA classification performance. Still, great accomplishment! The authors do not comment on the low performance for "no vocals" types of labels, it would be interesting to address it (my guess would be that this is a "negation" label which has huge support and variance and is hard to classify from 6sec clips). Considering contribution 2), it's a good replication of the work performed in the visual domain by several other groups (cited appropriately). The techniques used are sound and SOTA (RSA + Searchlight), but I do not have the knowledge to assess the specificities of the fMRI analysis. The baseline models used for comparison are sound and a welcome addition. It would have been helpful to provide the RSA matrices of the fMRI data itself, in order to visually compare to Figure 3, as is usually done in the visual DNN-RSA literature (if I missed it in the published papers I apologize, but couldn't find it in [23]). Given the strong ambient/symphonic vs rest subdivision, regressing that effect out would have been interesting (similar to [13] for animate vs inanimate). I'm not sure how to interpret the pretty low mean representational similarity in Figure 5A when going posterior along the STG. This doesn't make me want to trust the significance of the (layer 6 ; posterior STG) correlation much. Similarly, the drop in similarity as the layer index increases in Figure 4 is a bit contrary to what people have found when looking at IT correlations... It would seem that these results only show that early layers processing are similar to STG, but not later ones. The paper is a bit lacking in interpretations and discussions on that aspect. Given what is known about Wernicke's area, I would have liked to see style-conditioned similarities: did music genres with vocals representations significantly differ between hemispheres? Overall, it seems like part 2) is less discussed and lacks a bit of interpretation and context to be really useful for NIPS-attendees. But overall I think it is a strong paper which will find its audience.

Confidence in this Review

2-Confident (read it all; understood it all reasonably well)


Reviewer 6

Summary

The authors compare the internal representations of various DNNs to fMRI recordings of human STG during a music tagging task. The DNNs used are variants of the AlexNet with 1D convolutional layers which receive either signals in time, frequency or both as input. The DNNs are trained to predict tags from this input on the MagnaTagaTune dataset. Neuroimaging data were takes from the Studyforrest dataset. Three different DNN architectures are tested, and various other alterations are made to the AlexNet architecture (lines 105-110) but it remains unclear what the effects of these alterations are, and whether they are necessary. The tagging performance of the DNNs looks good, with apparently non-significant differences between the architectural choices. The analyses proceed with the DNN-TF model only (time and frequency input) because it showed the highest performance of any single model. The RDMs in fig. 3 indicate that deeper layers of the DNN cluster music into ambient/symphony and the rest. Next, RSA analyses of models vs. brain are presented. The first analysis reveals that the highest representational similarity between the _first_ convolutional layer and the brain (fig 4). In the second analysis, STG voxel clusters are compared to DNN layers. fig 5b shows that anterior STG correlates mostly with level 3, which is surprising given the results in fig. 4, which show the strongest correlation with layer 1 -- unless I misinterpreted the color coding, and fig 5b actually shows a mixture of colors between layers 1 and 5 (green and yellow). The authors claim that fig. 5 is evidence of a "representational gradient" (line 243), since the layer assignments increase from anterior to posterior. However, given the correlations in fig 5a, I don't see that. The correlations of the more posterior regions ( = higher layers ) seem to be virtually zero, so there is appears to be no measurable gradient. If the authors would like to make such a claim, I would require a demonstration that these tiny correlations of the deeper layers/posterior regions is still above chance, especially since their control models show correlations of roughly the same magnitude as these deeper layer, as figure 7 seems to indicate.

Qualitative Assessment

In summary, I think this manuscript presents an interesting line of work, but needs more elaboration before presenting it to the NIPS audience. In particular, as detailed in the summary, I am not convinced that the work shows the existence of representational gradients, as explained in the summary. further comments/questions: - line 67-68:"...top 50 tags...to ensure there is enough training data for each tag". How much training data per tag do you need? Since the DNN is a shared representation between tags, it should be possible to learn tag associations from a fairly small number of tag examples? - line 162: "multiplying them by the discrete cosine transform matrix" Why? Does this affect the dissimilarities? - How did you choose the architectures of the DNN models? Have you experimented with other choices? - line 173: "...but with parameters drawn from a zero mean and unit variance multivariate Gaussian..." It would be a more convicing control model if the parameters were drawn from the marginal statistics of the trained DNN, to reduce the chance that the differences you see are just due to differences in the mean predictions.

Confidence in this Review

3-Expert (read the paper in detail, know the area, quite certain of my opinion)